# FEDEVALFAIR: A Privacy-Preserving and Statistically Grounded Federated Fairness Evaluation Framework

## ABSTRACT

Federated learning has rapidly gained attention in the industrial sector due to its significant advantages in protecting privacy. However, ensuring the fairness of federated learning models post-deployment presents a challenge in practical applications. Given that clients typically rely on limited private datasets to assess model fairness, this constrains their ability to make accurate judgments about the fairness of the model. To address this issue, we propose an innovative evaluation framework, FEDEVALFAIR, which integrates private data from multiple clients to comprehensively assess the fairness of models in actual deployment without compromising data privacy. Firstly, FEDEVALFAIR draws on the concept of federated learning to achieve a comprehensive assessment while protecting privacy. Secondly, based on the statistical concept of 'estimating the population from the sample', FEDEVALFAIR is capable of estimating the fairness performance of the model in real-world settings from a limited data sample. Thirdly, we have designed a flexible two-stage evaluation strategy based on statistical hypothesis testing. We verified the theoretical performance and sensitivity to fairness variations of FEDEVALFAIR using Monte Carlo simulations, demonstrating the superior performance of its two-stage evaluation strategy. Additionally, we validated the effectiveness of the FEDEVALFAIR method on real-world datasets, including UCI Adult and eICU, and demonstrated its stability in dealing with real-world data distribution changes compared to traditional evaluation methods.

## KEYWORDS

Federated Learning, Model Fairness, Privacy, Federated Evaluation Framework, Statistical Inference, Hypothesis Testing

## 1 INTRODUCTION

Due to its significant advantages in protecting privacy, federated learning has quickly attracted the attention of the industrial sector. As the deployment of federated learning models in critical areas such as criminal sentencing, employee recruitment, and loan approval continues to expand, the issue of fairness in federated learning models has garnered widespread concern[1, 10]. Especially in recent years, many studies have introduced various concepts of fairness, such as group fairness. This concept aims to reduce biases against protected groups defined by sensitive attributes such as gender and race, and has become a focus of research[6, 23]. Group fairness is particularly concerned with mitigating the unfairness that may be exhibited by models during the training process, especially when models are trained on biased data. Although many studies have been devoted to measuring and mitigating this algorithmic bias, formal guarantees for fairness attributes are still lacking in practice[5, 8, 38, 40].

Ignoring the fairness evaluation of models during their actual deployment can lead to serious consequences[12, 19, 22, 36]. For instance, in models predicting patients' reactions to specific medications, if the differences in patients' genetic backgrounds, lifestyle habits, and environmental factors are not adequately considered, the model may produce misleading predictions for certain groups. Such biases, if not identified and corrected through reliable fairness evaluations, could lead to serious errors in treatment decisions for these patients, causing irreversible impacts on their health. However, there are problems with the methods of evaluating model fairness at the time of actual deployment. For example, existing methods for assessing model fairness are often based on existing datasets, but these datasets may not fully reflect the situation when the model is deployed in the real world, leading to these assessment methods failing to accurately reflect the model's fairness during actual deployment. Secondly, model developers often only have access to limited real-world data, and conducting more detailed fairness evaluations would require more data. However, considerations of data privacy limit the acquisition of additional data, further exacerbating the difficulty of accurately assessing model fairness.

As awareness of data privacy protection strengthens, obtaining private data to assess the fairness of machine learning models has become increasingly difficult and costly[11]. Many data providers, especially organizations dealing with sensitive information such as medical institutions and financial companies, are becoming more reluctant to share their data, fearing that this may violate privacy protection regulations or compromise the privacy and security of their client[31]. This situation poses a significant challenge for developers who aim to enhance model fairness and ensure that their decision-making processes do not adversely affect specific groups. They are faced with the problem of how to effectively assess model fairness in the absence of sufficient data support. Therefore, there is an urgent need to develop a new type of model fairness assessment method that can utilize multi-source private data for evaluation without infringing on data privacy.

Existing common methods for assessing model fairness primarily rely on evaluations conducted on test datasets, reflecting the model's fairness through metrics such as demographic parity or equality of opportunity[9, 26, 27]. However, a key limitation of this approach is its assumption that the test dataset can accurately represent the real-world data distribution when the model is deployed, an assumption that does not hold true in many practical scenarios. Therefore, model fairness assessments based solely on test datasets may not be reliable, especially when evaluating the fairness of models in real-world environments[5]. Given this, there is an urgent need to develop a new assessment method that can more accurately and reliably infer a model's fairness performance in actual deployment environments based on limited real-world datasets. Such a method would help more authentically capture and address potential fairness issues in models as they are applied in practice.

Under the premise of data privacy protection, especially when data does not leave the local environment, how to utilize the private

data of multiple participants to more reliably assess the fairness of models in actual deployment environments has become an important research question. To address this, we propose a model fairness federated evaluation framework named FedEvalFair, inspired by the concept of federated learning, aimed at enabling more reliable assessments of model fairness without direct access to the data. Specifically, the FedEvalFair framework first utilizes the bootstrap method at each participant's local site to generate multiple test sets through resampling. Then, based on the statistical principle of estimating population parameters from samples, it reliably estimates the model's fairness performance across different data providers. Moreover, we have developed an innovative two-stage fairness testing method, which can comprehensively assess the fairness of models on multi-source private data, thereby more accurately evaluating the model's fairness performance in actual deployment environments.

The FedEvalFair framework holds significant practical value. Relying solely on limited private data from a single source for model evaluation can lead to uncertainty and bias. FedEvalFair addresses this by enabling participation of multi-clients' datasets through privacy-preserving measures, substantially expanding the scope and depth of evaluation. Grounded in statistical principles like estimating population parameters from samples and hypothesis testing, it ensures reliable and accurate fairness assessment in real-world scenarios. Applicable not only for pre-deployment global fairness analysis of federated models to identify biases against disadvantaged groups, FedEvalFair can also periodically monitor fairness changes in dynamic data settings, facilitating timely model adjustments or retraining.

The main contributions of this study can be summarized as follows:

- We developed the first multi-source evaluation system dedicated to assessing model fairness, the FedEvalFair framework. Its core objective is to accurately evaluate the fairness of models when deployed in real environments, ensuring that the decision-making process is fair to all user groups.
- We proposed an innovative privacy-preserving framework that utilizes multi-party private datasets to comprehensively assess the fairness of models while fully protecting the data privacy of all participants. FedEvalFair ensures data security while allowing for in-depth analysis of model fairness.
- Drawing on the principles of estimating population parameters from samples and statistical hypothesis testing methods in statistics, we constructed a novel method for evaluating model fairness. This method enhances the reliability of evaluation results and provides a solid theoretical foundation for the quantitative analysis of model fairness.
- Compared to traditional methods confined to test datasets, FedEvalFair can more comprehensively assess model fairness in actual deployment environments. Experimental results demonstrate higher stability and reliability of our method when evaluating model fairness in real-world settings.

This paper is structured as follows: Section 2 reviews related work, setting the theoretical background for the FedEvalFair framework. Section 3 explains the framework's background, including assumptions and definitions. Section 4 details FedEvalFair's core, the two-stage algorithm, covering theoretical underpinnings and implementation. Section 5 presents experimental setup and results, showcasing the framework's effectiveness. Finally, Section 6 summarizes the findings and conclusions of this study.

## 2 RELATED WORK

### 2.1 Model Fairness Optimization

In the field of machine learning, the core concept of fairness is to ensure that models do not engage in discriminatory behavior when processing individuals or groups with different protected attributes, such as race, gender, age, and religion. These protected attributes, as part of the input features, should not alone determine the decision-making process of the model. As the understanding of model fairness in the machine learning field deepens, numerous studies have been dedicated to exploring methods to enhance fairness. Zafar et al. developed an innovative mechanism by introducing a fairness metric for decision boundaries, designing algorithm classifiers that can maintain both high accuracy and fairness, successfully applied to logistic regression and support vector machines[36]. Zhang et al. introduced a framework that reduces bias by training a predictor and an adversary model, with the goal of maintaining accuracy from input to prediction while preventing the adversary model from accurately predicting protected variables[38]. Additionally, with the increasing demand for privacy protection, researchers have also explored how to improve model fairness while protecting privacy. Ezzeldin et al.'s FairFed, an innovative server-side algorithm for fairness-aware aggregation in federated learning, aims to enhance group fairness without needing centralized access to sensitive data, showing the potential to build fairer models[18]. Yue et al.'s GIFAIR-FL framework, by adding regularization terms in federated learning to achieve both group and individual fairness, has proven to maintain or enhance predictive accuracy while improving fairness in image classification and text prediction tasks[35]. Zeng et al. introduced the FEDFB algorithm, which significantly improves model fairness by modifying the FEDAVG protocol to simulate centralized fair learning, achieving the performance of centrally trained models in some cases[37]. Du et al.'s AgnosticFair framework, aimed at achieving model fairness under unknown testing data distributions, achieves high accuracy and fairness on unknown testing data by kernel reweighting of training samples in the loss function and fairness constraints[16]. Mohri et al. proposed a concept of weak fairness for machine learning models, reducing bias during the training process by minimizing the maximum training loss for protected categories, but this approach is limited to the training stage and cannot ensure comprehensive fairness of the model[28].

### 2.2 Model Fairness Evaluation

Researchers have developed a variety of model fairness assessment strategies to identify unfair phenomena in machine learning models[1, 20, 40]. Biswas et al. proposed a causal fairness approach aimed at assessing the impact of data preprocessing stages on fairness in the machine learning process, demonstrating through case studies how to mitigate bias by selecting appropriate data transformers[6]. Joymallya Chakraborty et al. introduced the Fairway method, which reduces bias in models by adjusting strategies

during the preprocessing and training stages[10]. This method emphasizes the impact of real-world biases on the fairness of AI systems and demonstrates its effectiveness in discovering and reducing model biases while minimizing the impact on predictive performance, recommending the integration of bias testing and mitigation into standard practices of machine learning software development. Hort et al. introduced Fairea, a technique for evaluating the variability in model behavior as a result of machine learning bias mitigation methods, and tested the effectiveness of 12 bias mitigation methods through a large-scale empirical study[23]. Aggarwal et al. developed a novel method for automatically generating test inputs to detect individual discrimination in machine learning models, combining symbolic execution with local interpretability to efficiently create test cases, showing higher efficacy in detecting individual discrimination compared to existing benchmarks[1]. Zhang et al.introduced an innovative and scalable approach for identifying instances of individual discrimination in deep neural networks (DNNs) by employing lightweight processes like gradient computation and clustering, significantly enhancing scalability over current state-of-the-art techniques[40].

Despite these advancements, there remains a lack of empirical research to measure and compare the fairness of ML models in practical applications and to analyze the specific impact of mitigation algorithms on model performance[5]. As the protection of privacy increases, there is a particular focus in trustworthy federated learning on whether federated learning models can be deployed in real-world scenarios, and evaluating whether models comply with fairness principles during actual deployment is an important evaluation goal.

## 3 PRELIMINARIES

In this section, we will present the foundational setup related to the FEDEVALFAIR framework and provide definitions pertinent to model fairness.

### 3.1 Definition of Model Fairness

In the domain of model fairness metrics, fairness is often divided into two core categories: individual fairness and group fairness. Individual fairness focuses on evaluating whether the model's predictions for similar pairs of samples are fair[17, 25]. On the other hand, group fairness concentrates on assessing the balance of the model's predictions for specific protected groups (e.g., "women" vs. "men")[7, 8, 15, 19, 22]. This study will focus on the measurement methods of group fairness.

In existing research, to measure fairness, numerous metrics have been proposed[3, 4, 14]. These metrics typically involve calculating the classification ratios for specific groups, such as the true positive rate and false positive rate (e.g., true positives, false positives), and assess fairness by comparing the differences in these rates across different groups. Consider a dataset $D$ containing $n$ samples, with true classification labels $Y$, the model's predicted classification labels $\hat{Y}$, and a sensitive attribute $A$. In this setting, $A = 1$ can represent a privileged group (such as males), while $A = 0$ represents a non-privileged group (such as females). For binary model outputs $\hat{Y}$ (and their corresponding labels $Y$), we assume $\hat{Y} = 1$ indicates a positive outcome. We adopts typical group fairness metrics[5, 20],

with a particular focus on Demographic Parity (DP) and Equal Opportunity Disparity (EOD).

$$\Delta DP = \left| P(\hat{Y} = 1 | A = 0) - P(\hat{Y} = 1 | A = 1) \right|, \quad (1)$$

$$\Delta EO = \left| P(\hat{Y} = 1 | A = 0, Y = 1) - P(\hat{Y} = 1 | A = 1, Y = 1) \right|. \quad (2)$$

We denote the total disparity between $\Delta DP$ (Demographic Parity disparity) or $\Delta EO$ (Equal Opportunity disparity) as $\Delta Dis$. A model can be considered to have achieved fairness when it completes training and satisfies the following condition:

$$\Delta Dis - \varepsilon \le 0,$$

where $\varepsilon$ represents a small perturbation value, introduced to ensure that the fairness assessment accounts for a certain degree of flexibility.

For fairness across multiple data sources, a trained model $h$ is considered to have achieved fairness on multiple data sources if it satisfies the following condition for each data source:

$$\Delta Dis_h(k) - \varepsilon_k \le 0, \forall k \in \{1, \dots, N\}.$$

When evaluating model fairness across multi-source private data, we measure the composite fairness performance using $\Delta Dis(k)$ for fairness disparity on each of the $K$ clients (represent a data source). We derive multiple Bootstrap test datasets ($n_i$) from each client to compute Bootstrap disparity value, represented as $Y_{i1}, Y_{i2}, \dots, Y_{in_i}$. These datasets, created via non-parametric Bootstrap from limited original data, lead to varying statistical distributions. It is reasonable to assume that these Bootstrap disparity value follow a normal distribution $N(\mu_i, \sigma_i^2)$, an assumption that becomes increasingly reliable after a sufficient number of Bootstrap iterations. Thus, we can collect a comprehensive set of fairness disparity data for each client. According to the law of large numbers, the validity of this assumption strengthens with the increase in dataset size, which is also supported by our empirical evidence.

REMARK 1. *FEDEVALFAIR is based on locally obtained bootstrap fairness disparity distributions from each client, rather than on the raw data, so there is no assumptions about the clients' raw data.*

For the fairness disparity data on the $i$-th client, we can calculate the sample mean and sample variance as follows:

$$\overline{Y}_i = \frac{1}{n_i} \sum_{j=1}^{n_i} Y_{ij}, S_i^2 = \frac{1}{n_i - 1} \sum_{j=1}^{n_i} (Y_{ij} - \overline{Y}_i)^2.$$

From this, we can estimate the population mean and variance of fairness disparitys for each client as follows:

$$\hat{\mu}_i = E\left( \frac{1}{n_i} \sum_{j=1}^{n_i} Y_{ij} \right) = \frac{1}{n_i} \sum_{j=1}^{n_i} E(Y_{ij}) = \overline{Y}_i;$$

$$D(\hat{\mu}_i) = \left( \frac{1}{n_i} \right)^2 D\left( \sum_{j=1}^{n_i} Y_{ij} \right) = \frac{\sigma_i^2}{n_i},$$

where $E(X)$ denotes the mathematical expectation, and $D(X)$ denotes the variance of variable $X$.

We set $\mu_i = \mu + \Delta\mu_i$, where $\mu_i$ represents the population mean of fairness disparity for the model on the $i$-th client, and $\Delta\mu_i$ represents

the difference from the common mean $\mu$ across all clients. The hypothesis testing problem of interest is as follows:

$$H_0 : \mu_1 = \mu_2 = \mu_3 = \ldots = \mu_k = 0 \quad \text{vs} \quad H_1 : \exists i, \mu_i \neq 0. \quad (3)$$

We can divide this hypothesis testing problem into two stages: The hypothesis testing problem for the first stage is:

$$H_0 : \mu_1 = \mu_2 = \mu_3 = \ldots = \mu_k \quad \text{vs} \quad H_1 : \exists i, j, \mu_i \neq \mu_j. \quad (4)$$

The second stage of our analysis involves a critical hypothesis testing problem, formulated as:

$$H_0 : \mu_i = 0 \quad \text{vs} \quad H_1 : \mu_i \neq 0, \quad \text{for} \quad i = 1, 2, \ldots, k. \quad (5)$$

This test is versatile; for instance, it can be independently applied when a company seeks to evaluate the fairness of its model on a specific dataset. More significantly, in scenarios aiming for a comprehensive fairness assessment through the integration of multi-source private data, the application of this two-stage hypothesis test proves to be especially crucial. Addressing these hypothesis testing challenges is central to our study.

### 3.2 Bootstrap Method

Bootstrap is a powerful statistical method, whose core idea involves simulating the process of sampling from the original sample by repeated resampling, used to estimate statistical characteristics of sample data[29, 32, 39]. The Bootstrap method is divided into two main forms: parametric Bootstrap and non-parametric Bootstrap methods. The lies in that the parametric Bootstrap generates new sample sets based on the known statistical distribution of the sample, whereas the non-parametric Bootstrap does not assume any specific data distribution and directly performs replacement resampling from the original dataset to construct new sample sets. The application of non-parametric Bootstrap is particularly important for those data types whose statistical distributions are difficult to precisely define, such as images or audio.[2, 33, 34]

In practice, suppose the original dataset is $X = \{X_1, X_2, ..., X_n\}$, where $X_i$ represents the $i$th observation, and $n$ is the total number of observations in the dataset. For each resampling, we construct a new sample set $X_b^* = \{X_{b1}^*, X_{b2}^*, ..., X_{bn}^*\}$, where $b$ indicates the current resampling index, $b = 1, 2, ..., B$, and $B$ is the total number of resampling times. Each $X_{bi}^*$ is randomly and with replacement chosen from the original dataset $X$. This means that for each $b$, every $X_{bi}^*$ is independently drawn from $\{X_1, X_2, ..., X_n\}$. This process is repeated $B$ times, generating $B$ different bootstrap sample sets $X_1^*, X_2^*, ..., X_B^*$.

In model evaluation, the non-parametric bootstrap provides an approach that can be used to more accurately evaluate models and quantify their uncertainty, especially in complex environments where the data distribution is unknown.

## 4 METHODOLOGY

### 4.1 Overview of FedEvalFair

As illustrated in Figure 1, the FedEvalFair framework integrates the core concept of federated learning and places a special emphasis on enhancing data privacy. Unlike traditional evaluation methods, FedEvalFair avoids the direct access requirement to real datasets. Within this framework, instead of sharing their original data and

model parameters, clients only need to share intermediate parameters related to the central aggregation for evaluation purposes. Additionally, we apply homomorphic encryption algorithms to encrypt the shared parameters. These measures render it impossible for semi-honest yet curious servers to infer any relevant information about the private data of clients, thereby safeguarding client data privacy.

Moreover, another innovative aspect of the FedEvalFair framework is its reliance on statistical theory—specifically, the ideas of estimating populations from samples and statistical hypothesis testing—to conduct more trustworthy assessments of model fairness. Data holders independently utilize a non-parametric bootstrap method to create multiple bootstrap test datasets, to as comprehensively as possible evaluate the model's fairness performance on their respective datasets. This method effectively uses limited sample data to infer the model's fairness performance in a broader population data space. Based on statistical hypothesis testing theory, model fairness undergoes a two-stage rigorous hypothesis testing. Another significant advantage of the FedEvalFair is that evaluating model fairness on multi-clients' private data sources requires at most three rounds of communication. This not only demonstrates the practicality of the FedEvalFair method but also proves its effectiveness.

In the following sections, we will elaborate on the two-stage fairness evaluation process using the FedEvalFair framework, focusing on the theoretical underpinnings of Algorithm 1 and 2.

### 4.2 Two Stage Fairness Evaluation Method

In this subsection, based on the principles of statistical hypothesis testing, we provide a rigorous theoretical derivation for the two-stage fairness evaluation method, along with a detailed description of their algorithmic implementation processes.

Next, we begin with the theoretical underpinnings of Algorithm 1. When the null hypothesis $H_0$ in (4) is true and $\sigma_i$ is known, drawing upon the concept of Graybill-Deal estimation[21], the estimator for the common mean $\mu$ is derived as

$$\hat{\mu} = \frac{\sum_{i=1}^{k} \frac{1}{D(\hat{\mu}_i | \sigma_i)} \hat{\mu}_i | \sigma_i}{\sum_{i=1}^{k} \frac{1}{D(\hat{\mu}_i | \sigma_i)}} = \frac{\sum_{i=1}^{k} \frac{n_i}{\sigma_i^2} \hat{\mu}_i | \sigma_i}{\sum_{i=1}^{k} \frac{n_i}{\sigma_i^2}} = \sum_{i=1}^{k} \omega_i \hat{\mu}_i | \sigma_i, \quad (6)$$

where $\omega_i = \dfrac{\frac{n_i}{\sigma_i^2}}{\sum_{i=1}^{k} \frac{n_i}{\sigma_i^2}}, \quad i = 1, 2, \cdots, k.$ Then, a natural statistic is given by

$$Z_i = \frac{\hat{\mu}_i - \hat{\mu}}{D(\hat{\mu}_i)} = \frac{n_i}{\sigma_i^2}\left(\hat{\mu}_i - \sum_{i=1}^{k} \omega_i \hat{\mu}_i\right). \quad (7)$$

Under the null hypothesis $H_0$ in (3), the mathematical expectation and variance of $Z_i$ are given by

$$E(Z_i) = \frac{n_i}{\sigma_i^2}\left(E(\hat{\mu}_i) - \sum_{i=1}^{k} \omega_i E(\hat{\mu}_i)\right) = 0,$$

$$D(Z_i) = D\left(\frac{n_i}{\sigma_i^2}(\hat{\mu}_i - \sum_{i=1}^{k} \omega_i \hat{\mu}_i)\right) = \left(\frac{n_i}{\sigma_i^2}\right)^2\left(\frac{\sigma_i^2}{n_i} - \frac{1}{\sum_{i=1}^{k} \frac{n_i}{\sigma_i^2}}\right). \quad (8)$$

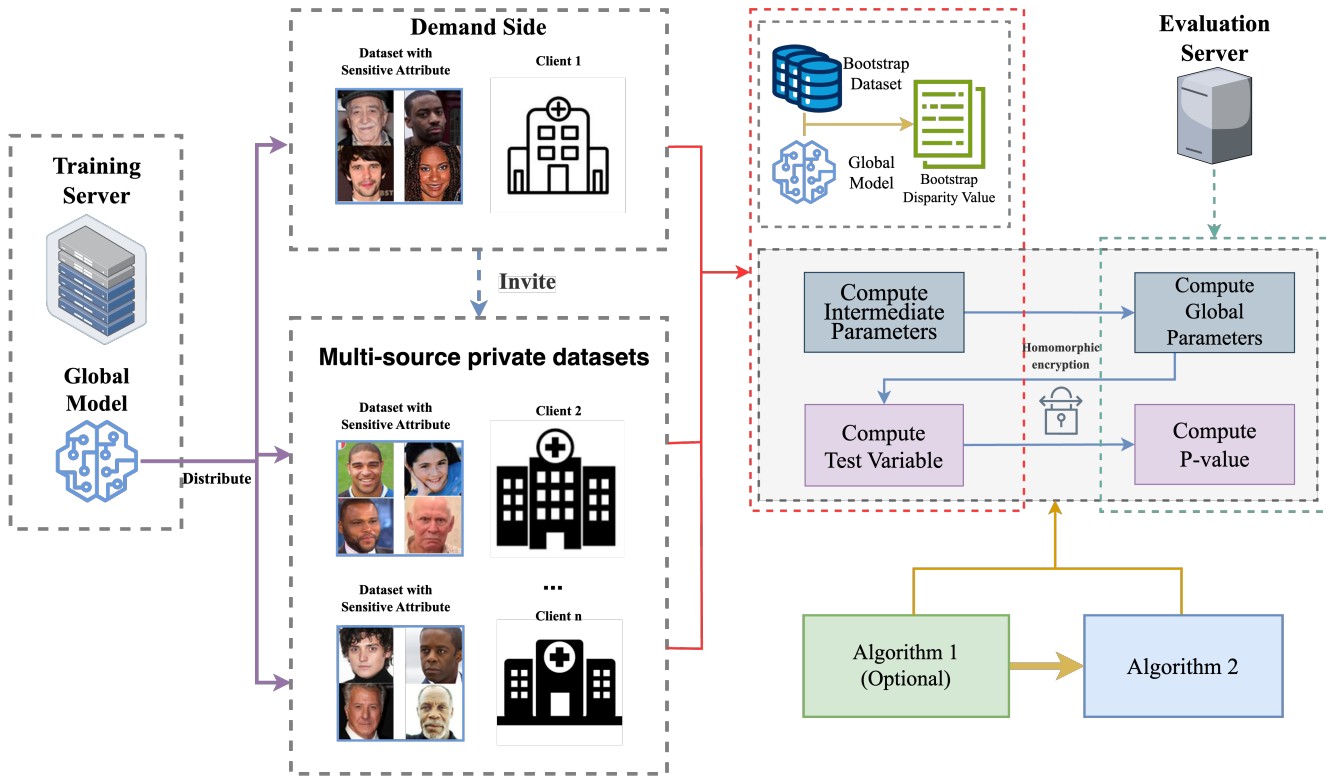

**Figure 1: Overview of the FedEvalFair Framework. This figure visually differentiates the architecture of the FedEvalFair framework from the traditional federated learning model, delineates the FedEvalFair framework, outlining its unique components and communication processes.**

By Central Limit Theorem, it readily follows that

$$\frac{Z_i - E(Z_i)}{\sqrt{D(Z_i)}} = \frac{\bar{Y}_i - \sum_{i=1}^{k} \omega_i \bar{Y}_i}{\sqrt{\frac{\sigma_i^2}{n_i}(1-\omega_i)}}, i = 1, 2, \cdots, k. \tag{9}$$

As $n_i \to \infty$, $\frac{Z_i - E(Z_i)}{\sqrt{D(Z_i)}}$ follows a standard normal distribution, denoted by $N(0,1)$, $i = 1, 2, \cdots, k$. Then, a natural statistic for testing (3) is given by

$$T = \sum_{i=1}^{k} \left( \frac{\bar{Y}_i - \sum_{i=1}^{k} \omega_i \bar{Y}_i}{\sqrt{\frac{\sigma_i^2}{n_i}(1-\omega_i)}} \right)^2 = \sum_{i=1}^{k} \frac{n_i}{\sigma_i^2} \frac{\left( \bar{Y}_i - \sum_{i=1}^{k} \omega_i \bar{Y}_i \right)^2}{1-\omega_i}.$$

Before delving into the detailed discussion of the test statistic $T$, we first introduce Lemma 1, which provides a theoretical underpinning for our analysis. Here, $\overset{asy}{\sim}$ is used to denote approximate adherence to a certain statistical distribution.

LEMMA 1. *Suppose* $X_j \overset{asy}{\sim} N(0,1)$, $j = 1, 2, \ldots, k$, *and* $X_1, X_2, \cdots, X_k$ *are mutually independent of each other, then* $\sum_{j=1}^{k} X_j^2 \overset{asy}{\sim} \chi^2(k)$.

PROOF. We provide a detailed proof of Lemma 1 in the Appendix. □

According to Lemma 1, we can infer $T \overset{asy}{\sim} \chi^2(k-1)$. In practice, $\sigma_i^2$ is often unknown. Therefore, we replace $\sigma_i^2$ with the sample variance $s_i^2$, $i = 1, 2, \cdots, k$, which enables us to obtain a new test statistic, i.e.,

$$T_1 = \sum_{i=1}^{k} \frac{n_i}{s_i^2} \frac{\left( \bar{Y}_i - \sum_{i=1}^{k} \hat{\omega}_i \bar{Y}_i \right)^2}{1-\hat{\omega}_i} = \sum_{i=1}^{k} \frac{n_i}{s_i^2} \frac{\left( \bar{Y}_i - \sum_{i=1}^{k} \frac{1}{\sum_{i=1}^{k} \frac{n_i}{s_i^2}} \frac{n_i}{s_i^2} \bar{Y}_i \right)^2}{1 - \frac{1}{\sum_{i=1}^{k} \frac{n_i}{s_i^2}} \frac{n_i}{s_i^2}}. \tag{10}$$

As a result, we can set up the test rule, as given by

$$p = P\left(T_1 > \chi^2_{k-1,\alpha}\right),$$

where $\chi^2_{k-1,\alpha}$ denotes the upper $\alpha$-th quantile of a chi-square distribution with degrees of freedom $df = k - 1$. The null hypothesis $H_0$ in (3) is rejected whenever the above $p$-value is less than the nominal significance level of $\alpha$, which indicates that at least two means are unequal. In the context of this study, $H_0$ is rejected if the model exhibits differing fairness disparities across at least two data sources.

We now present Algorithm 1 that operationalizes the concepts discussed above.

**Algorithm 1** The First Stage of FedEvalFair

1: **Input:**Data of $k$ clients
2: **for** each client (in parallel) **do**
3:    Clients compute $\overline{Y}_i, S_i^2, S_i^2/n_i$
4:    Clients encrypt $\overline{Y}_i, S_i^2, S_i^2/n_i$ and upload to the server
5: **end for**
6: Server computes $\hat{\mu}, \sum_{i=1}^{k} n_i/S_i^2$ on the EncryptedData and distributes to all clients
7: **for** each client (in parallel) **do**
8:    Clients decrypt $\hat{\mu}, \sum_{i=1}^{k} n_i/S_i^2$, and compute $\omega_i, Z_i$, and $(Z_i - E(Z_i))/\sqrt{D(Z_i)}$
9:    Clients encrypt $\omega_i, Z_i, (Z_i - E(Z_i))/\sqrt{D(Z_i)}$ and upload to the server
10: **end for**
11: Server computes $T_1$ on the EncryptedData
12: Server computes the $p$-value
13: **Output:** $p$-value

Next, we present the theoretical foundation for the second phase of fairness evaluation, which entails devising a solution to the hypothesis testing problem (5). Subsequently, we will design a test statistic specifically for this hypothesis testing. Based on the Central Limit Theorem, we can deduce the following:

$$W = \sqrt{\sum_{i=1}^{k} \frac{n_i}{\sigma_i^2}} (\hat{\mu} - \mu) = \sqrt{\sum_{i=1}^{k} \frac{n_i}{\sigma_i^2}} \left( \frac{\sum_{i=1}^{k} \frac{n_i}{\sigma_i^2} \hat{\mu}_i}{\sum_{i=1}^{k} \frac{n_i}{\sigma_i^2}} - \mu \right).$$

In the context of hypothesis testing problem (5), when the null hypothesis holds true, we obtain the test statistic as follows:

$$W^* = \sqrt{\sum_{i=1}^{k} \frac{n_i}{\sigma_i^2}} \left( \frac{\sum_{i=1}^{k} \frac{n_i}{\sigma_i^2} \hat{\mu}_i}{\sum_{i=1}^{k} \frac{n_i}{\sigma_i^2}} - \mu_0 \right).$$

If $\sigma_i^2$ is known, then $W^*$ serves as the test statistic for hypothesis testing problem (10). However, in practical scenarios, $\sigma_i^2$ is often unknown. Therefore, we can substitute the maximum likelihood estimate of $\sigma_i^2$ into $W^*$, yielding the following result:

$$W^* = \sqrt{\sum_{i=1}^{k} \frac{n_i}{s_i^2}} \left( \frac{\sum_{i=1}^{k} \frac{n_i}{s_i^2} \overline{Y}_i}{\sum_{i=1}^{k} \frac{n_i}{s_i^2}} - \mu_0 \right).$$

It is evident that as $n_i \to \infty$ for $i = 1, 2, ..., k$, it follows that $W^* \sim N(0, 1)$. Subsequently, we can establish the testing rule as follows:

$$p = 2 \times \min \left\{ P(U > w^*), P(U < w^*) \right\}.$$

where $w^*$ is the observed value of $W$, and $U$ represents a random variable following the standard normal distribution $N(0, 1)$. The null hypothesis $H_0$ in (10) is rejected if the p-value is less than the nominal significance level of $\alpha$. This rejection indicates that: the model evaluation results on these distributed private data sources

are unfair, implying that the model, when deployed in practice, exhibits bias towards certain groups with sensitive attributes.

Building upon the theoretical framework, Algorithm 2 translates these ideas into actionable steps for implementation.

**Algorithm 2** The Second Stage of FedEvalFair

1: **Input:**Data of $k$ clients
2: **for** each client (in parallel) **do**
3:    Clients compute $\overline{Y}_i, S_i^2, S_i^2/n_i$
4:    Clients encrypt $\overline{Y}_i, S_i^2, S_i^2/n_i$ and upload to the server
5: **end for**
6: Server computes $\sum_{i=1}^{k} n_i/S_i^2$ on the EncryptedData and distributes to all clients
7: **for** each client (in parallel) **do**
8:    Clients decrypt $\hat{\mu}, \sum_{i=1}^{k} n_i/S_i^2$ and compute $\omega_i, \omega_i \overline{Y}_i$
9:    Clients encrypt $\omega_i, \omega_i \overline{Y}_i$, and upload to the server
10: **end for**
11: Server computes $w^*$ on the EncryptedData
12: Server computes the $p$-value
13: **Output:** $p$-value

## 5 EXPERIMENTS

In this section, we first analyze the theoretical performance of FedEvalFair and its sensitivity to fairness perception through Monte Carlo simulation experiments. Subsequently, we validate the effectiveness of FedEvalFair on real datasets. Finally, we design a comparative experiment to test the stability of FedEvalFair in real-world scenarios dealing with non-independent and identically distributed (non-IID) data, and compare its reliability with that of traditional evaluation methods.

### 5.1 Experiment Setup

*5.1.1 Datasets and Models.* We utilized the Adult[28] and eICU datasets[30] for federated learning training with the Logistic Regression model. For the Adult dataset, which includes over 40,000 records of adults, we employed the settings within the [13], segregated the data into two clients: one encompassing individuals with a doctoral degree, and the other consisting of those without. During this process, we identified race (with a focus on White individuals) as a sensitive attribute, with the objective of predicting whether an individual's annual income surpasses $50,000. The training spanned 1800 rounds, with learning rates adjusted between 0.02 and 0.10. For the eICU dataset, this dataset contains over 200,000 samples, covering 17 different features. our attention was directed towards the patient data table, which detailed ICU admissions across various hospitals. This dataset was divided into multiple groups, with each group representing a consolidation of data from six hospitals, amounting to a total of 11 clients[24]. In this context, African American was designated as the sensitive attribute, aiming to forecast whether a patient's hospital stay exceeded one week. This training comprised 2300 rounds, with a consistent learning rate of 0.50.

In the Monte Carlo simulations, we set a nominal significance level of 5% and carried out 5000 iterations. Similarly, we configured a wide range of parameter combinations to simulate the performance of the fairness assessment methods in different real-world scenarios. For details on parameter settings, please see Appendix and our GitHub: https://github.com/anonymous5929/FedEvalFair.

All experiments are conducted on Ubuntu 22.04.3 server with 48 cores of 2.20GHz CPU, 256GB RAM, and NVIDIA V100 GPUs with 32 GB memory.

## 5.2 Experiment Results

*5.2.1 Monte Carlo Simulation.* In the Monte Carlo simulation experiments, we focused on evaluating the effectiveness and sensitivity of the FedEvalFair method in detecting unfairness. Since FedEvalFair is built upon the theoretical foundation of statistical hypothesis testing, its effectiveness is typically assessed by the Type I error rate. If the Type I error rate is close to the set significance level, it indicates that the likelihood of FedEvalFair mistakenly categorizing a fair model as unfair is extremely low. Regarding sensitivity to detecting unfairness, we measure this through the power of the FedEvalFair method. The higher the power, the more effectively the FedEvalFair method can identify unfair situations. Therefore, the experiments aim to validate that FedEvalFair maintains a low Type I error rate while demonstrating high sensitivity and detection capability for unfairness.

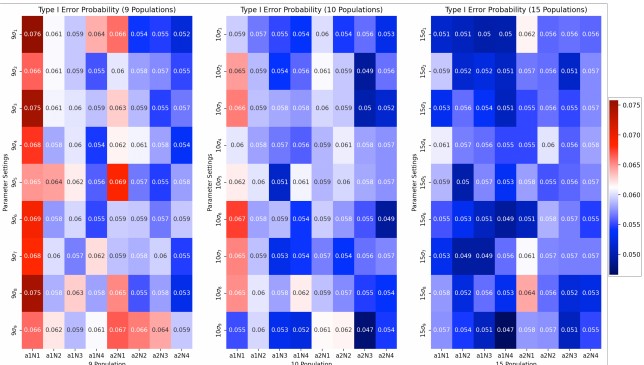

**Figure 2: Probability of Type I error for Algorithms 1-2 through Monte Carlo simulations. When the value is around 0.05, it indicates that at a significance level setting of 5%, Algorithms 1-2 are almost unlikely to make errors.**

As shown in Figure 2, in the vast majority of scenarios, both Algorithm 1 and 2 effectively maintain the Type I error probability under the 5% nominal significance level. Notably, when the sample size is small, the probability of a Type I error may sometimes be slightly higher than 0.05. However, as the sample size increases, this error probability rapidly decreases towards 5%. This phenomenon indicates that while the performance of these algorithms may fluctuate under small sample conditions, they can stably control the Type I error at an ideal level with larger sample sizes.

Figure 3 demonstrates the effectiveness of the FedEvalFair method in identifying unfairness, where an increase in efficacy indicates a stronger ability to recognize unfairness. As the fairness deviation increases, both Algorithm 1 and 2 show a significant improvement in efficacy, proving that FedEvalFair can quickly and accurately detect fairness disparity among clients and within the model. Notably, this detection capability significantly enhances with an increase in sample size, highlighting FedEvalFair's potential in monitoring model fairness and preventing fairness drift.

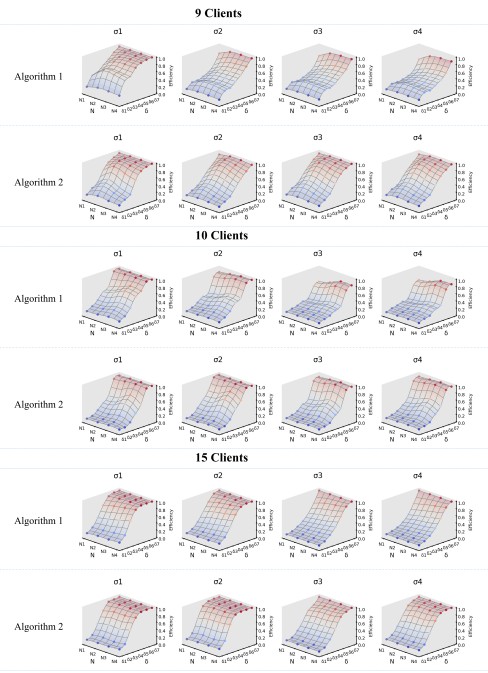

**Figure 3: Power performance of Algorithm 1-2 in detecting varying degrees of unfairness through Monte Carlo simulations. As the degree of unfairness increases, the power of both Algorithm 1 and 2 increases rapidly, indicating the effectiveness and sensitivity in detecting unfairness.**

*5.2.2 Experiments on Real-Datasets.* Next, we turn our attention to experiments conducted on real datasets. Our main goal is to verify the effectiveness of the FedEvalFair method in real datasets. Specifically, we will explore key questions: First, when a model actually maintains fairness across multiple distributed data sources, can the FedEvalFair method accurately recognize and confirm this? Second, when a model is actually unfair, can the FedEvalFair method effectively give a rejection conclusion and accurately identify unfair situations? To carry out our experiments, we referenced the FCFL algorithm proposed by Cui et al.[13]. This algorithm employs a multi-objective optimization strategy to train a fair federated learning model and notes that models trained without fairness constraints tend to be unfair. Based on this theory, we used the FCFL algorithm to simulate fair and unfair models on the Adult and eICU datasets.

Firstly, consider the simulated unfair model. On the Adult and eICU datasets, the evaluation results based on Algorithm 2 showed that the p-value is far less than 0.0001. This indicates that the disparity value of the model is significantly different from 0. Similarly, we performed non-parametric bootstrap on the test set to mimic the data distribution in real-world scenarios as closely as possible. As illustrated in Figure 4, the bootstrap disparity value of the model is also significantly different from 0, which corroborates the conclusion of the FedEvalFair. This suggests that if such a model is deployed in a real production environment by the data provider, it may exhibit bias against groups with sensitive attributes.

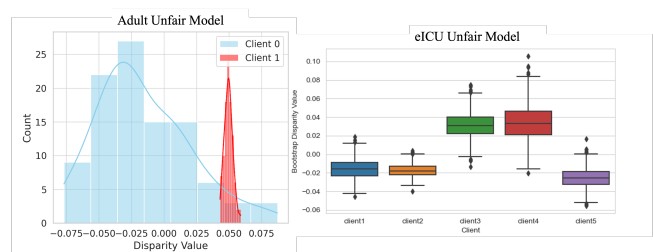

**Figure 4: The Bootstrap disparity distribution plot of the simulated unfair model on real datasets shows significant statistical differences in fairness performance across different clients. Additionally, the values of DP and EO are significantly different from 0. This indicates that the model is unfair overall.**

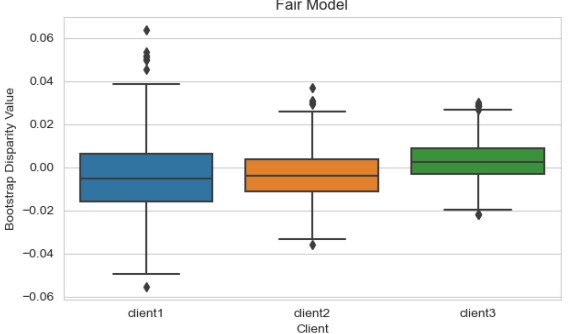

**Figure 5: The Bootstrap disparity value of the simulated fair model based on FCFL. As indicated by the Bootstrap disparity values in the graph, the fairness disparities of this model are nearly close to 0 on every client.**

Next, for the simulated fair models, Figure 5 displays the model's bootstrap disparity value across multiple private data sources. The results suggest these disparity values are essentially zero, allowing us to infer the model is fair. Based on the FedEvalFair method's comprehensive fairness evaluation of this model on these private data sources, the p-values are **0.1523** (Algorithm 1) and **0.2256** (Algorithm 2), above the 0.05 significance level. This indicates the model is statistically considered fair, meeting the fairness requirements for deployment in real production environments.

We designed a new comparative experiment aimed at evaluating the reliability of the FedEvalFair method compared to existing evaluation methods (EEM) when assessing model fairness in real deployment environments. By summarizing existing research on fairness in federated learning([28],[37],[13],etc), existing evaluation methods typically calculate the model's fairness disparity values based solely on test datasets, assuming that the test dataset can represent the data distribution encountered in actual deployments. However, limited test datasets often fail to fully reflect the data distribution in real environments. While existing methods can compare the strengths and weaknesses of different algorithms, they may lack accuracy when evaluating model fairness in actual deployments. In contrast, the FedEvalFair method adopts the concept of "estimating the population from the sample," aiming to infer the

model's performance on real data as accurately as possible from limited datasets, thus better reflecting the model's fairness performance in production environments. For experimental details and the experimental framework, please refer to the appendix.

Specifically, we designed a comparative experimental framework mainly to evaluate the stability of the EEM and the FedEvalFair method on partitioned test datasets and simulated real datasets. We considered various comparison scenarios: *1. When the distribution of the test dataset is completely consistent with the real production dataset; 2. When there is a varying degree of shift between the test dataset and the real production dataset; 3. When the test dataset and the real production dataset are completely different distributions.* We used Cohen's d value to calculate the difference in results between the two evaluation methods under the test dataset and the real production dataset, reflecting whether there is a significant difference between the two sets of values. The larger the Cohen's d value, the more significant the difference, and a Cohen's d value greater than 0.8 indicates a very significant difference.

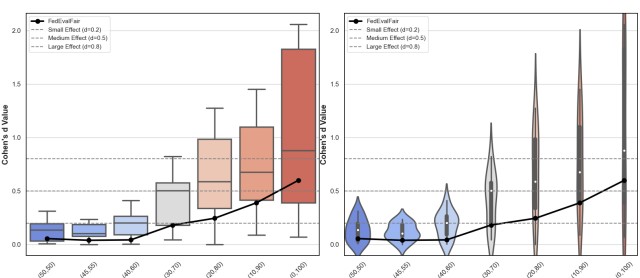

**Figure 6: Evaluation Stability Comparison Between EEM and FedEvalFair**

**Table 1: Cohen's d values for different methods**

| cohen | (50, 50) | (30, 70) | (20, 80) | (10, 90) |
|---|---|---|---|---|
| EEM | 0.1503 | 0.5558 | 0.9650 | 1.0880 |
| FedEvalFair | **0.0546** | **0.1814** | **0.2447** | **0.3895** |

As shown in Figure 6 and Table 1, EEM already exhibits a Cohen's d value exceeding 0.8 when the distribution shift between the test dataset and the real production dataset is greater than a certain threshold, indicating a very significant difference. In contrast, while the FedEvalFair method also shows some difference, its stability remains superior to that of EEM. This finding suggests that the FedEvalFair framework has excellent reliability in assessing the fairness of models when deployed in real production environments.

## 6  CONCLUSION

In this paper, we introduce FedEvalFair, a new framework that leverages statistical sampling and hypothesis testing principles along with multi-source private data to assess fairness in real-world model deployments, ensuring data privacy. By employing bootstrapping, FedEvalFair accurately evaluates fairness, utilizing a flexible two-stage strategy. This framework not only provides theoretical backing for federated learning applications but also promotes the advancement of trustworthy federated learning.

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
