# OpenReview forum: "FedEvalFair: A Privacy-Preserving and Statistically Grounded Federated Fairness Evaluation Framework"
_acmmm.org/ACMMM/2024/Conference — MM2024 Poster_

### Official Review · Reviewer_1xLS · 2024-05-25

**Rating:** 4
**Confidence:** 1

**Summary:**

The paper introduces FedEvalFair, a novel evaluation framework designed to assess the fairness of federated learning models in real-world deployments while preserving privacy. It addresses the challenge of limited and biased fairness assessments resulting from reliance on test datasets. By integrating multi-client data and leveraging federated learning principles, FedEvalFair offers a comprehensive fairness evaluation without compromising data privacy. The framework incorporates statistical sampling methods and a two-stage evaluation strategy based on hypothesis testing, demonstrated through Monte Carlo simulations to provide reliable fairness analysis. Validation on datasets like Adult and eICU confirms its effectiveness. This work constitutes the first multi-source evaluation system dedicated to assessing model fairness, enhancing fairness measurement accuracy.

**Strengths:**

The paper proposes an innovative framework, FedEvalFair, aimed at solving the problem of fairness evaluation after the deployment of federated learning models, ensuring that model decisions are fair to all user groups while strictly protecting data privacy. This framework integrates private data from multiple parties, and combines statistical sampling theory and hypothesis testing methods to achieve a comprehensive evaluation of model fairness without compromising data privacy. Its core advantage lies in providing a two-stage evaluation strategy, which has been validated for effectiveness and stability through Monte Carlo simulation and on real-world datasets such as Adult and eICU.

For the ACM MM community, although the experiments in this paper focus on fairness evaluation of single-mode data, the framework proposed is expected to be applied in the future to federated learning and fairness assurance of multimodal data in the multimedia field.

**Limitations:**

The FedEvalFair proposed in the paper is novel, but the explanation of the paper is not clear.

1)	The paper does not explicitly mention which homomorphic encryption algorithm it adopts

2)	In the face of imbalanced classification, such as when a client lacks data about a certain classification (such as women), How FedEvalFair calculates DP

3)	Why divide the algorithm into two stages. These two stages do not seem to have a clear sequential relationship and can be merged into one stage.

4)	What does the arrow from Algorithm 1 to Algorithm 2 mean in Figure 1? It seems that Algorithm 2 does not rely on the results generated by Algorithm 1.

5)	The description of Figure 3 is unclear, especially regarding the missing explanation of the two coordinates at the bottom of the chart, which makes understanding difficult. This critical information should be presented in the paper rather than in supplementary materials.

6)	The description of Figure 6 is unclear. What does the horizontal axis (50,50) mean? What do different colors represent?

7)	This paper summarizes the EEM based on three relatively earlier papers (with the most recent one being only in 2021), without comparing FedEvalFair with more and recent methods.

**Suitability:**

2

---

### Official Review · Reviewer_Vzd8 · 2024-05-27

**Rating:** 4
**Confidence:** 3

**Summary:**

This paper explores the fairness of federal learning, and the authors propose a model fair federal assessment framework named FedEvalFair. By adopting the concept of federated learning, the proposed framework is able to assess model fairness more reliably without direct access to data. In order to improve the operability, the authors developed a two-stage fairness test method to comprehensively evaluate the fairness performance of the model in a real-world deployment environment. In addition, abundant experiments and visual analysis are carried out to verify the effectiveness of the proposed framework.

**Strengths:**

1. This paper proposes a novel research question and develop the first multi-source evaluation system specifically designed to evaluate model fairness. Ensure that the decision process is fair for all user groups by evaluating the fairness of the model at deployment time.
2. This paper propose a new principle of sample estimation of population parameters and the method of statistical hypothesis testing, a method of evaluating model fairness is designed. The quantitative analysis strategy constructed by this method is helpful to the experimental analysis of model fairness.

**Limitations:**

1. The study did not analyze links to the multimedia community or the work's contribution to research in the multimedia field. The author should clearly state in the paper the significance, value, or potential impact of the research on the multimedia field.
2. Authors should specify which algorithms the baselinse in Table 1 are refer to, including reference information or links to sources.
3. There are some spelling mistakes in the paper (e.g., page 5, line 521).
4. In the writing of the paper, some formulas are missing labels (e.g., page 3, lines 301 and 309, etc.)

**Suitability:**

2

---

### Official Review · Reviewer_Myeo · 2024-06-01

**Rating:** 2
**Confidence:** 3

**Summary:**

This paper proposes FedEvalFair, a privacy-preserving and statistically grounded federated fairness evaluation framework for assessing the fairness of machine learning models post-deployment. The proposed framework enables the evaluation of model fairness using multi-source private data without compromising data privacy, through the use of federated learning and statistical methods.

**Strengths:**

FedEvalFair is stated to be the first multi-source evaluation system dedicated to assessing model fairness. It introduces a novel approach to evaluating the fairness of models in real-world deployment environments.

**Limitations:**

1. The motivation behind the framework seems unclear.
- If the framework is intended for use in federated learning (FL) settings, it would be more appropriate to integrate it directly into the FL process to assess model fairness across various training rounds during the analysis and evaluation phase.
- Conversely, if the framework is designed as a general tool for evaluating fairness among multiple clients, it falls short by not considering the practical FL factors such as client participation rate, data heterogeneity across clients and etc.
2. The related work section lacks thoroughness. The paper provides an overview of previous studies without clearly comparing their respective strengths, weaknesses, and specific application scopes. It would be better to divide the previous works into several categories and separately introduce their application scope and advantages.
3. The writing quality needs enhancement.
- In Section 3.1, the paper only presens the mathematical formulations of DP and EOD without offering insights into their meanings, derivation processes or distinctions.
- The paper fails to articulate its improvement made to the Bootstrap method, especially in the FL scenario.
4. The evaluation of the framework is insufficiently. The paper does not benchmark its performance against existing fairness evaluation methods in FL.
5. The use of hypothesis testing to assess fairness is not adequately supported by existing references. The authors are expected to clairy its superiority to existing works on fairness evaluation.
6. The generalizability of the proposed method to other metrics is uncertain, which could constrain the overall impact of this work.
7. The evaluation is based on simplistic models and datasets, which may not adequately demonstrate the practical applicability of the proposed method.

**Suitability:**

2

---

### Meta-Review · Area_Chair_eaZb · 2024-07-01

**Recommendation:** Accept (Poster)
**Confidence:** 5

**Metareview:**

The paper presents FedEvalFair, a new evaluation framework that assesses the fairness of federated learning models in real-world deployments while maintaining privacy. It tackles the issue of limited and biased fairness assessments caused by relying on test datasets. I believe this paper will provide valuable insights into privacy-preserving techniques in federated learning, and the proposed evaluation framework can stimulate further research in this area. Therefore, I recommend accepting this paper.